# Lip-Text Verification Using Multivariate Time Series Lip Motion Features

Turi Abu & Polina Konovalova
Time Series Intelligence Course Project Proposal

April 28, 2026

**Abstract**

Lipreading has recently gained attention not only as a complementary modality to audio-based speech processing, but also as a promising tool for security-oriented applications such as identity verification, liveness detection, and audio-visual consistency checking. By analyzing the visual patterns of a speaker's lip movements, these systems can help determine whether spoken content matches the observed facial motion, making them particularly useful in scenarios where audio may be spoofed, manipulated, or unavailable. Early research in lipreading relied on handcrafted visual features, including pixel intensities, geometric lip contours, and Active Appearance Models (AAMs). More recent approaches, however, leverage deep learning techniques to learn representations directly from raw video data. While these methods have demonstrated strong performance, they often depend on large-scale datasets and tend to produce high-dimensional, less interpretable feature representations. In this proposal, we investigate an alternative formulation of lipreading for verification tasks. Specifically, we propose to model lip-text verification as a multivariate time series learning problem using compact geometric features extracted from facial landmarks. Instead of processing raw images, we derive a low-dimensional representation that captures the temporal dynamics of lip articulation. These features are designed to be interpretable, computationally efficient, and robust to speaker variation through normalization. We propose a hybrid deep learning architecture combining temporal convolution and recurrent sequence modeling to learn discriminative patterns from the extracted time series. The system is designed to verify whether a given lip motion sequence corresponds to a claimed text sequence. This proposal outlines the feature design, modeling approach, and experimental plan for evaluating the effectiveness of this representation.

## 1 Introduction

### 1.1 Motivation

Visual Speech Recognition (VSR), commonly referred to as lipreading (or lip reading), aims to infer spoken content from visual cues of a speaker's facial movements. It has been extensively explored as a complementary modality to audio-based speech recognition (Audio-Visual Speech Recognition, AVSR), particularly in scenarios where acoustic signals are noisy, corrupted, or unavailable. Early studies in human perception have shown that visual information from lip movements significantly contributes to speech intelligibility, especially under adverse listening conditions [1].

One of more prominent applications of audio-based voice recognition within security context is verification in biometric systems. While the natural assumption in security AVSR is to treat every input as a unique identifier, recent work on speaker verification systems highlights the potential for Time-Domain Voice Identity Morphing (TD-VIM): signal-level morphed voice inputs that demonstrate high capability to match characteristics of multiple identities and bypass biometric verification [2]. VSR remains a strong security solution against TD-VIM attacks, interacting with visible articulatory motions and facial landmarks to infer spoken content and verify its contents, highlighting the importance of VSR research and analysis for security deployments.

### 1.2 Gaps Identified

Traditional approaches to VSR relied on handcrafted visual features extracted either from pixel intensities or from parametric models of facial shape and appearance. Methods based on Discrete Cosine Transform

(DCT), Principal Component Analysis (PCA), and Active Appearance Models (AAMs) demonstrated that both geometric and appearance-based features could encode useful speech information [3, 4]. In particular, comparative studies have shown that appearance features, which capture pixel-level information inside the mouth region, are generally more informative than shape-based features alone [5]. However, such representations are often high-dimensional and sensitive to input quality and overall imaging conditions, such as lighting, resolution, and speaker variability.

Recent advances in deep learning have paved for a significant improvement of VSR performance by learning representations directly from raw video data. End-to-end models based on spatiotemporal convolutions and sequence modeling, such as LipNet [6] and large-scale transformer-based architectures [7], have achieved strong results on benchmark datasets. Despite their demonstrated success, these approaches typically rely on a vast corpus of labeled data and availability substantial computational resources, and produce representations that are difficult to interpret.

## 1.3 Contribution

In this work, we propose a lip-text verification approach that models lip motion as a multivariate time series using compact geometric features derived from facial landmarks. By focusing on temporal dynamics rather than raw pixel data, the method provides a low-dimensional, interpretable, and efficient representation. This formulation enables robust modeling under limited data while leveraging time series techniques to capture articulatory patterns for verification.

# 2 Literature Review

## 2.1 Visual Speech Recognition

Early VSR systems relied on hand-crafted visual features such as Discrete Cosine Transform (DCT) coefficients of mouth region images or Active Appearance Models (AAMs) [4]. With the advent of deep learning, end-to-end approaches dominate: LipNet [6] introduced sequence-to-sequence lip reading using spatiotemporal convolutions and recurrent layers; LRW [8] and LRS benchmarks [9] fueled development of transformer-based architectures [7, 10]. However, these methods require large-scale datasets and GPU-intensive training pipelines.

## 2.2 Landmark-Based Approaches

An alternative line of work uses facial landmarks as intermediate representations. Lan et al.[5] demonstrated that geometric features from lip contours can achieve competitive accuracy for isolated word recognition. Fernandez-Lopez and Sukno [11] surveyed landmark-based methods and noted their robustness to illumination and speaker variation. Our work extends this direction by formulating the extracted features explicitly as multivariate time series and applying temporal modeling techniques from the time series analysis literature.

## 2.3 Lip Reading for Identity Verification

Lip reading has emerged as a promising modality for identity verification, leveraging unique behavioral and physiological characteristics of lip movements during speech. It is contactless, difficult to spoof, and inherently supports liveness detection when combined with random prompts [12]. Recent work focuses on disentangling lip motion into semantic content, static appearance, and behavioral dynamics to improve robustness [12, 13]. Beyond camera-based methods, wireless sensing offers a privacy-preserving alternative. BackLip [14] leverages Wi-Fi backscatter signals, analyzing energy distribution and steady-state characteristics of lip-induced signal variations for lip-reading base user authentication.

## 2.4 Time Series Classification

Time series classification (TSC) has been extensively studied [15]. Deep learning approaches include Fully Convolutional Networks (FCN) [16], 1D ResNets [17], and attention-based models [18]. Hybrid architectures combining convolutional layers for local feature extraction with recurrent layers for temporal modeling have proven effective for variable-length multivariate time series [19, 20]. Our model follows this hybrid paradigm, combining 1D-CNN with BiGRU.

# 3    Objectives

The objective of this work is to develop a lip-text verification system that leverages compact geometric features extracted from facial landmarks and models them as multivariate time series to capture temporal dynamics of lip motion. The approach aims to balance performance, interpretability, and computational efficiency by using low-dimensional, scale-invariant representations and a hybrid temporal modeling framework. The goal is to build a system that can reliably verify the consistency between lip motion and claimed text while remaining robust under limited data and practical deployment conditions.

# 4    Proposed Methodology

## 4.1    Feature Extraction

From the detected facial landmarks, we construct a multivariate time series representation of lip motion. Each video frame is mapped to a 5-dimensional feature vector, resulting in a sequence $\mathbf{X} \in \mathbb{R}^{T \times 5}$, where $T$ denotes the temporal dimension.

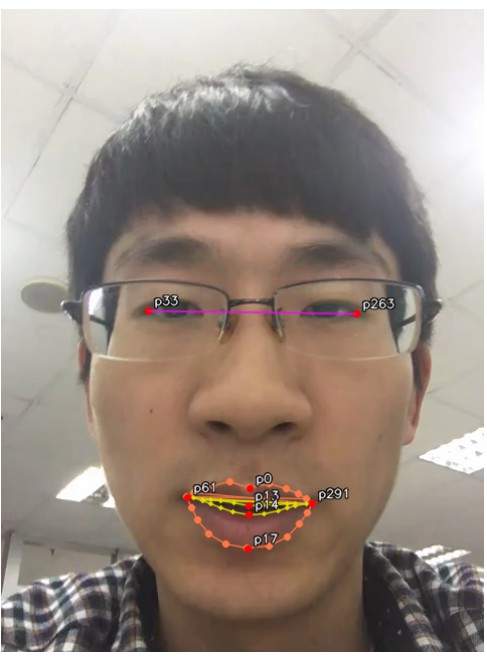

Figure 1: Face landmarks being used to calculate the lip motions features including inter-ocular distance

To account for speaker-specific factors such as head size, all features are normalized using the inter-ocular distance computed as the Euclidean distance between the outer corners of the eyes.

$$d_{\text{eye}} = \|\mathbf{p}_{33} - \mathbf{p}_{263}\|. \tag{1}$$

Distance-based features are divided by $d_{\text{eye}}$, while area-based features are normalized by $d_{\text{eye}}^2$. This normalization ensures scale invariance and improves cross-speaker generalization. We extracted 5 features from geometry of lip as time series.

1. **Vertical Aperture**

   The first feature captures the vertical aperture of the mouth, defined as the normalized distance between the upper and lower inner lip landmarks:

   $$f_1 = \frac{\|\mathbf{p}_{13} - \mathbf{p}_{14}\|}{d_{\text{eye}}}. \tag{2}$$

   This feature measures mouth opening and correlates strongly with phonetic categories, distinguishing closed-mouth consonants from open vowels.

2. **Horizontal Spread**

   The second feature measures horizontal spread, defined as the normalized distance between the left and right mouth corners:

   $$f_2 = \frac{\|\mathbf{p}_{61} - \mathbf{p}_{291}\|}{d_{\text{eye}}}. \tag{3}$$

   This captures lip stretching and rounding, which are important articulatory cues for differentiating vowel classes.

3. **Inner Lip Area**

   Inner lip area is computed using the Shoelace formula over the ordered inner lip landmarks:

   $$f_3 = \frac{A_{\text{inner}}}{d_{\text{eye}}^2}, \tag{4}$$

   where the polygon area is given by

   $$A_{\text{inner}} = \frac{1}{2} \left| \sum_{i=0}^{n-1} (x_i y_{i+1} - x_{i+1} y_i) \right|. \tag{5}$$

   It captures the area of the actual oral opening. Directly related to the visible oral cavity. More discriminative than outer lip area for distinguishing speech sounds because it ignores lip thickness variations and focuses on the true opening.

4. **Compactness (Roundness)**

   Compactness is defined as

   $$f_4 = \frac{4\pi \cdot A_{\text{outer}}}{P_{\text{outer}}^2}, \tag{6}$$

   where $A_{\text{outer}}$ and $P_{\text{outer}}$ denote the area and perimeter of the outer lip contour, respectively. This quantity reflects the circularity of the lip shape and distinguishes between rounded and spread articulations.

5. **Lip Speed**

   The fifth feature captures lip motion dynamics through the magnitude of velocity in the geometric feature space:

   $$f_5 = \sqrt{\left(\frac{df_1}{dt}\right)^2 + \left(\frac{df_2}{dt}\right)^2}. \tag{7}$$

   This feature encodes the speed of articulatory transitions, which is particularly informative for distinguishing rapid consonant movements from sustained vowel articulations.

## 4.2 Model

The model takes as input a time series lip feature sequence $\mathbf{X} \in \mathbb{R}^{B \times T \times C}$ and a vector of claimed digit labels $\mathbf{d} \in \mathbb{Z}^B$, where $B$ denotes the batch size, $T$ the number of frames per segment, and $C$ the dimensionality of the lip features. The sequence $\mathbf{X}$ is first processed by a shared LipEncoder, which consists of two temporal one-dimensional convolutional layers followed by a bidirectional gated recurrent unit (BiGRU) [21]. The convolutional layers capture local articulatory dynamics (e.g., rapid mouth opening and closing), while the BiGRU models long-range temporal dependencies across the time series sequence.

To handle variable-length segments, a binary mask is applied over the temporal dimension, enabling masked mean pooling over the BiGRU outputs. This operation produces a fixed-dimensional embedding vector $\mathbf{e}^{\text{lip}} \in \mathbb{R}^{B \times D}$, where $D$ denotes the embedding dimension. In parallel, each claimed digit label is mapped to a learned embedding $\mathbf{e}^{\text{digit}} \in \mathbb{R}^{B \times D}$ via an embedding lookup table spanning all $N$ digit classes.

The resulting embeddings are concatenated along the feature dimension to form a joint representation:

$$[\mathbf{e}^{\text{lip}} \,\|\, \mathbf{e}^{\text{digit}}] \in \mathbb{R}^{B \times 2D}.$$

This joint vector is then passed through a two-layer classifier composed of a linear projection, a ReLU activation, dropout for regularization, and a final linear layer producing a scalar logit for each sample. The logit reflects the model's confidence that the observed lip motion corresponds to the claimed digit.

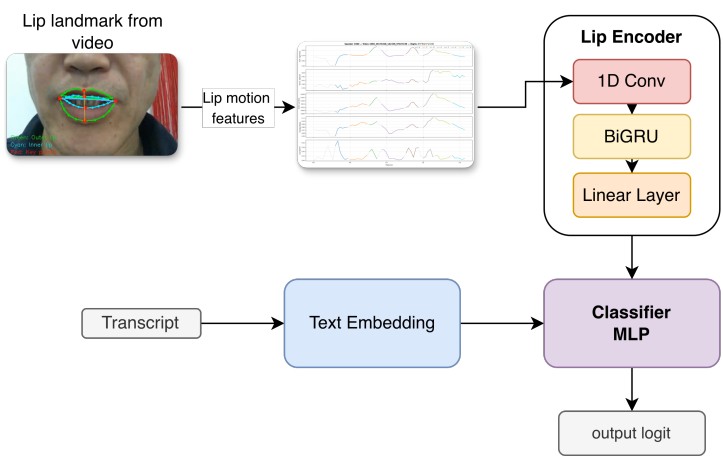

Figure 2: Proposed model architecture

## 4.3 Dataset

The dataset consists of short video recordings of speakers reading aloud sequences of eight digits in Chinese (e.g., "3 5 3 9 6 7 8 7"). All videos are captured using the front-facing (selfie) camera of users' mobile phones.

Each video is accompanied by temporal annotations obtained by ASR model trained for digit recognition from the audio:

$$
\begin{aligned}
\text{Digit sequence:} \quad & 2\ 6\ 8\ 0\ 9\ 2\ 9\ 8 \\
\text{Timestamps (s):} \quad & 0.57\text{–}0.75,\ 0.78\text{–}0.90,\ 0.93\text{–}1.11,\ 1.14\text{–}1.32, \\
& 1.35\text{–}1.56,\ 1.59\text{–}1.71,\ 1.74\text{–}1.89,\ 1.92\text{–}2.34
\end{aligned}
$$

## 5 Progress

So far, we have completed data preprocessing and established a baseline through initial experiments using the proposed time-series representation of lip motion features, baseline result presented in Appendix B. The current setup includes the construction of positive and negative training pairs and a basic verification model trained under this framework. The baseline demonstrates that the approach is feasible.

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

# A  Project schedule

Table 1: Project Timeline and Task Distribution

| Week | Dates | Tasks | Details | Lead / Support |
|---|---|---|---|---|
| 7 | Apr 17 – Apr 20 | Background study | Literature review on lip reading and time series classification | Turi, Polina |
| 8 | Apr 21 – Apr 24 | Proposed method and architecture | Data preprocessing and training baseline model | Turi, Polina |
| 9 | Apr 25 – Apr 28 | Proposal | Proposal writing and PPT preparation | Turi, Polina |
| 10 | Apr 29 – May 5 | Model improvement | Experiment with alternative architectures (e.g., deeper CNN, GRU/LSTM variants) | Turi, Polina |
| 11 | May 6 – May 12 | Feature refinement | Improve feature design and normalization, analyze feature importance | Turi, Polina |
| 12 | May 13 – May 19 | Training optimization | Hyperparameter tuning and training strategy improvements | Turi, Polina |
| 13 | May 20 – May 26 | Evaluation | Extensive evaluation, error analysis, and robustness testing | Turi, Polina |
| 14 | May 27 – Jun 2 | System refinement | Final model selection and performance improvement | Turi, Polina |
| 15 | Jun 3 – Jun 9 | Report writing | Writing final report and preparing ppt | Turi, Polina |
| 16 | Jun 10 | Presentation | Final project presentation | Turi, Polina |

# B  Preliminary Experiment

So far, we have completed data preprocessing and conducted preliminary experiments to establish a baseline. The model was trained for 100 epochs using the proposed time-series representation of lip motion features.

## B.1  Pair Generation

To train the verification model, we construct both positive and negative pairs. For each positive pair (i.e., a correctly matched lip sequence and digit label), we generate one negative pair using two strategies. First, in sequence shuffling, the digit order is randomly permuted while keeping the corresponding lip segments unchanged. Second, in digit replacement, 2–4 positions in the digit sequence are replaced with incorrect digits. These strategies introduce hard negative samples and encourage the model to learn meaningful alignment between lip motion and digit labels.

## B.2 Loss Function

We use Binary Cross-Entropy with Logits as the training objective:

$$\mathcal{L} = -\frac{1}{N} \sum_{i=1}^{N} \left[ y_i \log \sigma(\hat{y}_i) + (1 - y_i) \log \left(1 - \sigma(\hat{y}_i)\right) \right], \tag{8}$$

where $y_i \in \{0, 1\}$ is the ground-truth label, $\hat{y}_i$ is the predicted logit, and $\sigma(\cdot)$ denotes the sigmoid function:

$$\sigma(x) = \frac{1}{1 + e^{-x}}. \tag{9}$$

## B.3 Evaluation Metrics

The model is evaluated using standard verification metrics. The Area Under the ROC Curve (AUC) measures the ability of the model to distinguish between positive and negative pairs across different thresholds. The Equal Error Rate (EER) is defined as the point where the False Acceptance Rate (FAR) equals the False Rejection Rate (FRR):

$$\text{EER} \; : \; \text{FAR}(\tau) = \text{FRR}(\tau), \tag{10}$$

where $\tau$ is the decision threshold. In addition, we report the classification accuracy computed at the EER threshold:

$$\text{Accuracy} = \frac{TP + TN}{TP + TN + FP + FN}. \tag{11}$$

## B.4 Results

The baseline model achieves an AUC of 0.777, an EER of 0.2885, and an accuracy of 0.712 at the EER threshold. These results indicate that the model captures useful temporal patterns from lip motion, while also highlighting room for further improvement.

# C Challenges

## C.1 Co-articulation Effect

One of the challenges in visual speech modeling is the co-articulation effect, where the articulation of a given unit is influenced by its surrounding context. Unlike isolated pronunciation, the visual realization of a digit depends not only on its identity but also on preceding and succeeding digits, leading to variability in lip motion patterns.

Figure 3 illustrates this phenomenon using per-digit time series features extracted from a single utterance of the digit sequence "9 7 0 2 7 2 3 0". Although the same digits appear multiple times (e.g., "7" and "0"), their corresponding feature trajectories differ noticeably across positions in the sequence. This variation arises because the transition dynamics between neighboring digits alter the shape, timing, and magnitude of lip movements.

Such variability poses a challenge for modeling, as the system cannot rely on a single fixed pattern for each digit. Instead, it must learn context-dependent representations that are robust to these temporal and articulatory variations. This observation, in turn, further motivates the use of sequential models capable of capturing temporal dependencies rather than treating each segment as an independent static pattern.

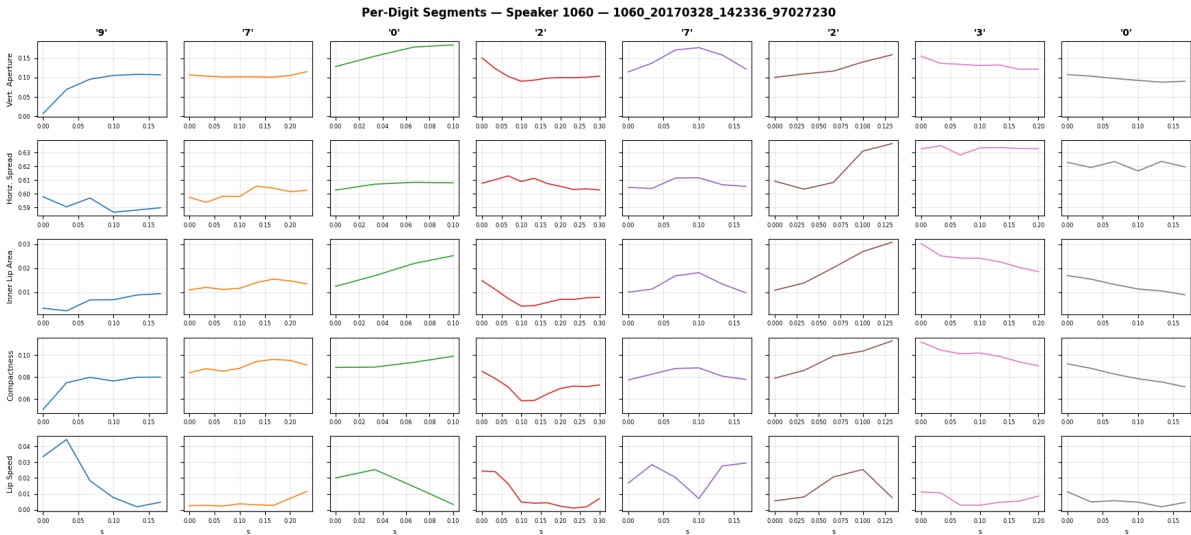

Figure 3: Per-digit lip motion time series for the sequence "9 7 0 2 7 2 3 0". Each column corresponds to a digit segment, and each row represents one of the extracted features. Repeated digits (e.g., "7" and "0") exhibit different temporal patterns depending on their position.

