# OpenReview forum: "Lip-Text Verification Using Multivariate Time Series Lip Motion Features"
_tsinghua.edu.cn/THU/2026/Spring/ANM — THU 2026 Spring ANM Submission_

### Official Review · Reviewer_e5vJ · 2026-05-12

**Rating:** 7
**Confidence:** 4

**Summary:**

This proposal presents a lip-text verification system that uses multivariate time series of geometric lip features (e.g., vertical aperture, horizontal spread, inner area, compactness, lip speed) extracted from facial landmarks. The authors propose a hybrid architecture combining 1D convolutions and a BiGRU to model temporal dynamics, paired with digit embeddings for claimed text verification. The work is motivated by security applications (e.g., anti-spoofing, liveness detection) and addresses gaps in interpretability and data efficiency compared to pixel-heavy deep learning methods. Preliminary results show AUC 0.777, EER 0.2885, and accuracy 0.712.

**Strengths:**

1. **Well-defined problem formulation:** Framing lip-text verification as a multivariate time series learning task is novel and appropriate for the course theme.

2. **Interpretable and efficient feature design:** The five geometric features are physically motivated, normalized (inter-ocular distance), and low-dimensional, which is a clear strength over black-box video models.

3. **Hybrid temporal architecture:** Combining 1D-CNN (local articulatory patterns) with BiGRU (long-range dependencies) is sensible and well-justified.

4. **Practical negative pair generation:** Two strategies (sequence shuffling and digit replacement) produce hard negatives, which is thoughtful for verification tasks.

5. **Acknowledgment of key challenge:** The co-articulation effect is explicitly discussed with an illustrative figure, showing domain awareness.

6. **Good project scope:** Feasible within a semester, with clear task distribution and timeline.

**Weaknesses:**

1. **Limited novelty in time series modeling:** The hybrid CNN+RNN approach is standard (e.g., LSTM-FCN from 2018). The proposal does not compare to or motivate why simpler TSC baselines (e.g., 1D ResNet, InceptionTime) might be inferior.

2. **Missing baseline comparisons:** Only one model result is reported. No comparison against non-temporal baselines (e.g., frame-average features + MLP) or other TSC methods.

3. **Small feature set (5 dimensions):** While interpretable, five features may lack discriminative power for fine-grained digit distinction. No analysis of feature redundancy or information content is provided.

**Questions:**

1. How many unique speakers and total video segments are in the dataset? Is there a speaker-disjoint evaluation split to test generalization?

2. How sensitive are the five features to head pose variation? The normalization only handles scale, not rotation or translation.

3. How was the digit embedding dimension chosen, and why concatenate rather than attend over the joint representation?

4. Will you compare against an end-to-end video model (e.g., LipNet on the same dataset) to quantify the trade-off between interpretability and accuracy?

---

### Official Review · Reviewer_Uaa9 · 2026-05-14

**Rating:** 8
**Confidence:** 4

**Summary:**

The authors propose a lip-text verification system that models lip motion as a multivariate time series using compact geometric features extracted from facial landmarks. Instead of using raw video frames, each frame is converted into five interpretable lip-motion features: vertical aperture, horizontal spread, inner lip area, compactness, and lip speed. These features are normalized by inter-ocular distance for scale invariance. A hybrid 1D-CNN and BiGRU model encodes the lip-motion sequence, combines it with an embedding of the claimed digit label, and predicts whether the observed lip movement matches the claimed text. The proposal includes a model architecture diagram, preliminary experiments, and an analysis of co-articulation effects.

**Strengths:**

- Clear and Interpretable Feature Design: The five proposed lip-motion features are well defined mathematically and are easy to interpret. The normalization by inter-ocular distance is also a good design choice for improving cross-speaker robustness.
- Strong Time Series Framing: The proposal successfully reformulates lip-text verification as a multivariate time series problem.
- Good Visualization: The proposal includes useful visualizations, especially the landmark figure and the model architecture diagram. The co-articulation figure in the appendix is also helpful because it demonstrates why repeated digits can have different trajectories depending on context.

**Weaknesses:**

* Dataset Description Is Incomplete: The proposal does not clearly state the number of speakers, number of videos, train/validation/test split, or whether the evaluation is speaker-dependent or speaker-independent. This is important because cross-speaker generalization is central to verification.
* Limited Text Scope: The current formulation uses digit labels, so the task is closer to digit-sequence verification than general lip-text verification. The title and claims should make this limitation clearer.
* Baseline Comparisons Are Missing: The proposal would be stronger with simple baselines such as with simple/traditional models. Without these, it will be harder to show that the proposed CNN-BiGRU architecture is necessary.

**Questions:**

- How many speakers and videos are included in the dataset, and are test speakers unseen during training?
- Is the final goal digit verification only, or will the method generalize to arbitrary text sequences?

---

### Official Review · Reviewer_4QkC · 2026-05-15

**Rating:** 8
**Confidence:** 4

**Summary:**

The paper proposes a lip-text verification system based on compact geometric features extracted from facial landmarks, modeled as multivariate time series. Rather than processing raw video frames end-to-end, the authors derive a 5-dimensional feature vector per frame capturing vertical aperture, horizontal spread, inner lip area, compactness, and lip speed, all normalized by inter-ocular distance. These sequences feed into a hybrid 1D-CNN + BiGRU architecture that jointly embeds the lip motion sequence and a claimed digit label to produce a verification score. The proposal includes a preliminary baseline achieving AUC 0.777 and EER 0.2885 on a self-collected Chinese digit dataset, and provides an honest discussion of the co-articulation challenge.

**Strengths:**

The feature design is the strongest part of this proposal. The five geometric features are well-motivated individually, with phonetic justifications for vertical aperture and horizontal spread grounded in articulatory phonetics, and the inter-ocular normalization is a principled choice for cross-speaker invariance. This is more thoughtful than proposals that either use raw pixels or treat landmark distances as black-box inputs.
The fact that the authors already have preliminary results is a significant plus. AUC 0.777 at proposal stage, on a self-collected dataset, with a basic architecture, is a credible starting point and demonstrates the pipeline is actually running end-to-end. The pair generation strategy using both sequence shuffling and digit replacement to construct hard negatives also shows careful thinking about the verification training setup.
The co-articulation discussion in Appendix C is very good. Figure 3 directly illustrates the problem with real data from their own pipeline, and the authors connect it back to their modeling choices rather than treating it as an afterthought.

**Weaknesses:**

The dataset is the weakest point of the proposal and receives almost no critical attention. The authors describe a self-collected dataset of mobile phone recordings with ASR-derived timestamps, but give no information about dataset size, number of speakers, recording conditions, or how ASR errors in the timestamp annotations are handled. For a verification task, the number of speakers directly determines whether the evaluation is meaningful, since a 5-speaker dataset and a 50-speaker dataset would lead to very different conclusions about generalization.
The negative pair construction is functional but potentially leaky in the verification framing. If all negatives are constructed by permuting or substituting within the same speaker's recording, the model may learn to detect superficial sequence-level mismatches rather than genuine lip-text inconsistency. It is not clear whether the evaluation includes cross-speaker negative pairs, which would be the harder and more security-relevant case.
The model architecture section describes a fairly standard 1D-CNN + BiGRU pipeline, which is fine, but the proposal gives no justification for specific design choices such as number of convolutional layers, kernel sizes, hidden dimensions, or embedding size.
Finally, the connection to the security application stated in the introduction, specifically defending against TD-VIM attacks, is entirely dropped after Section 1. The proposal reads as two separate pieces: a security motivation and a time series classification system. The final paper will need to bridge these more explicitly, or the motivation should be scaled back to match what is actually being evaluated.

**Questions:**

How many speakers and recordings does the dataset contain?
Are negative pairs constructed across speakers, or only within the same speaker?
The timestamp annotations come from an ASR model. What is the word error rate of that model on this data, and how do you handle cases where the ASR-derived segmentation is incorrect?
Do you plan to evaluate on unseen speakers only, or is the test set mixed?

---

### Official Review · Reviewer_7ZKe · 2026-05-15

**Rating:** 7
**Confidence:** 4

**Summary:**

This proposal frames lip-text verification as a multivariate time series classification task. Five geometric features (vertical aperture, horizontal spread, inner lip area, compactness, lip speed) are extracted from facial landmarks, normalized by inter-ocular distance, and fed into a hybrid 1D CNN and bidirectional GRU conditioned on digit label embeddings. A custom mobile-phone dataset of Chinese digit sequences is used, with a baseline achieving AUC=0.777, EER=0.2885, and accuracy=0.712.

**Strengths:**

The time series formulation is a refreshing departure from end-to-end video frame approaches. By compressing lip motion into a compact geometric feature space, the authors trade some raw representational power for interpretability and efficiency, which is a meaningful design choice for security applications. The connection to TD-VIM attacks and audio-visual consistency checking gives the work a timely practical angle, and digit sequences provide a pragmatic scope for a semester project. The feature set captures complementary aspects of articulation, and the inter-ocular normalization is a sensible preprocessing step. Architecturally, the hybrid CNN-RNN design with digit label conditioning is appropriate for this task. The authors have also made solid practical progress: preprocessing is complete, a working baseline exists, hard negatives have been mined, and evaluation protocols are defined.

**Weaknesses:**

The baseline performance is a significant concern. An equal error rate of 28.85% means the system fails on nearly three in ten attempts, and 71.2% accuracy at threshold is barely above chance for a balanced task. The authors should frankly acknowledge this gap and define concrete targets, such as EER below 10% or AUC above 0.95, that would indicate success. Several critical experimental details are missing: the number of speakers, the train-validation-test split, whether evaluation is cross-speaker or speaker-dependent, and the balance of positive versus negative pairs. Without this information, it is impossible to tell whether the poor EER reflects a solvable modeling problem or a fundamental data limitation. The five-feature representation may also be too reductive; protrusion, asymmetry, and tongue visibility are absent, and given the high error rate, expanding the feature set deserves exploration. Co-articulation across 8-digit sequences may exceed the receptive field of the proposed two-layer CNN and modest BiGRU, so the authors should consider whether dilated convolutions or attention mechanisms are needed. Finally, the evaluation is limited to Chinese digits, which constrains both linguistic diversity and practical applicability; robustness to speaking rate, head pose, and lighting should at least be discussed.

**Questions:**

How many speakers are in the dataset, and is evaluation cross-speaker or speaker-dependent?

What target EER or AUC would you consider a successful outcome?

Have you explored additional geometric features such as lip protrusion or asymmetry?

How do you plan to handle variable speaking rates?

Have you tried alternative fusion strategies for lip and digit embeddings?

---

### Official Review · Reviewer_Urj5 · 2026-05-16

**Rating:** 9
**Confidence:** 4

**Summary:**

The proposed research aims to enhance identity verification and liveness detection by ensuring consistency between a speaker’s lip movements and the text they claim to be reciting. Rather than relying on high-dimensional raw video data, which can be computationally expensive and difficult to interpret, the authors extract five specific geometric features from facial landmarks: vertical aperture, horizontal spread, inner lip area, compactness, and lip speed. These features are normalized based on inter-ocular distance to account for variations in head size and camera proximity. The core architecture utilizes a 1D-CNN to capture localized articulatory dynamics combined with a Bi-directional Gated Recurrent Unit (BiGRU) to model long-range temporal dependencies in the speech sequence. The system is evaluated on a dataset of mobile phone videos where subjects read 8-digit Chinese sequences, using temporal annotations derived from an audio-based automatic speech recognition (ASR) model.

**Strengths:**

A primary strength of this approach is its interpretability and efficiency. By reducing the input to five compact geometric features, the model avoids the "curse of dimensionality" associated with raw video while remaining grounded in actual human speech mechanics. The normalization process further ensures the model's robustness against speaker-specific physical differences or varying recording distances. Additionally, the focus on "Time-Domain Voice Identity Morphing" provides a clear, high-stakes application for the research in the field of security and spoofing prevention.

**Weaknesses:**

One significant weakness lies in the challenge of co-articulation, as the visual pattern of a digit changes depending on the digits surrounding it, making it difficult for the system to rely on fixed patterns for each number. Also, the current performance, highlighted by an accuracy of 71.2% and an Equal Error Rate of 0.2885, suggests that the system may require further refinement before it is reliable enough for commercial security deployment. There is also a heavy dependency on the accuracy of the audio-based ASR used for labeling, which could introduce errors into the training process if the audio quality is not good enough.

**Questions:**

Since you noted that digit patterns differ based on their position in the sequence, do you plan to use any specific context-aware mechanisms beyond the BiGRU to handle these variations?

You utilize five specific geometric features. Have you conducted an ablation study to see if some (like Inner Lip Area) are significantly more discriminative than others (like Compactness) for digit recognition?

The dataset uses front-facing selfie cameras. How do you expect the model to handle significant head movement or changes in lighting that might happen if a user is moving around?

If the system is used for "liveness detection," how would it handle a "replay attack" where a high-resolution video of a user is played back to the camera?

---

### Official Review · Reviewer_8ooo · 2026-05-18

**Rating:** 8
**Confidence:** 3

**Summary:**

The authors propose a lip-text verification system that frames lipreading as a multivariate time series problem. Rather than processing raw video pixels they extract 5 geometric features per frame from facial landmarks: vertical mouth aperture, horizontal spread, inner lip area, compactness, and lip motion speed. All features are normalized by interocular distance for cross-speaker robustness. The features feed a hybrid model (two 1D convolutional layers followed by a bidirectional GRU), which is combined with a learned embedding of the claimed digit sequence to produce a binary match/no-match score. A preliminary baseline on a self-collected dataset on Chinese digit-sequence videos achieves AUC 0.777, EER 0.2885, and accuracy 0.712.

**Strengths:**

- The feature design is the strongest part of this proposal. Each of the 5 features relates to a real articulatory phenomenon (vowel openness, lip spreading vs rounding, oral cavity size, lip shape, articulatory speed), and the inter-ocular normalization is a sensible fix for cross-speaker variation. The math is fully specified, formulas for each feature and for the verification loss are present, and the proposal is precise enough that someone could reimplement it basically from scratch.
- Having a working baseline already running at proposal stage is a real plus. It shows the pipeline is end-to-end and gives the authors something concrete to iterate from. The negative pair construction (sequence shuffling and digit replacement) is also thoughtful for a verification setup.
-The co-articulation analysis in Appendix C is good, it illustrates the problem on the authors' own data and ties it back to the modeling choice, rather than treating it as a side remark.

**Weaknesses:**

- The dataset description is incomplete. The proposal does not state the number of speakers, the total number of videos, the data split, or whether evaluation is speaker-dependent or speaker-independent. A 5-speaker setup and a 50-speaker setup would lead to very different conclusions and generalization.
- The baseline performance is modest. EER 0.29 means the system gets it wrong almost three times out of ten, which is not yet at the level needed for the security framing in Section 1. The proposal would be stronger with a concrete target (e.g., EER below 0.10) trying to close the gap.
There is no comparison with simpler baselines. A small end-to-end video model would give a reference point for whether the CNN+BiGRU is necessary. The proposal also limits itself to Chinese digits, which is fine for scope but a limitation for generalization.

**Questions:**

-How many speakers are in the dataset, and is the test set speaker-disjoint?
-What EER or AUC would you consider a successful outcome?
-What is the error rate of the ASR model used for label generation, and how do you handle cases where the segmentation is wrong?

---

### Official Review · Reviewer_1Dr2 · 2026-05-18

**Rating:** 5
**Confidence:** 4

**Summary:**

[AI Review] This paper proposes a lip-text verification system using multivariate time series lip motion features. The review identifies critical issues including fundamental confusion between identity verification (who is speaking) and content verification (what was said), lack of speaker-disjoint evaluation that undermines result validity, and circularity in using audio ASR for boundaries while motivating visual-only verification. The method shows marginal novelty over prior landmark-based approaches, and the 'time series' framing is not leveraged through any standard temporal methods (DTW, shapelets, wavelets). Baseline performance is poor (AUC=0.777, EER=0.2885) and the 1:1 negative ratio is inadequate for verification tasks. With addressed fixes, the predicted score could reach 6-7/10.

**Strengths:**

1. The paper addresses a relevant problem in lip-text verification with a clear application motivation.
2. The use of multivariate lip motion features for verification is a reasonable and potentially useful approach.
3. The authors provide quantitative evaluation metrics (AUC and EER) for their proposed method.
4. The research direction has potential value if properly scoped as content verification rather than identity verification.

**Weaknesses:**

1. Critical conceptual confusion between identity verification and content verification persists throughout the paper, undermining the entire framing and evaluation of the work (Severity 9).
2. No speaker-disjoint evaluation is provided, meaning reported results may simply reflect learning speaker identity rather than genuine lip-text correspondence (Severity 9).
3. ASR circularity exists in the methodology — the system uses audio ASR to determine phonetic boundaries but is motivated as a visual-only verification solution (Severity 8).
4. Marginal novelty over Lan et al. [5] landmark-based features; the contribution does not clearly advance beyond prior work (Severity 8).
5. The 'time series' framing is forced and unused — no DTW, shapelets, or wavelet methods are employed despite this being the paper's claimed angle.
6. TD-VIM security framing mentioned in introduction is never connected to the proposed method.
7. Baseline performance is poor for a verification task (AUC=0.777, EER=0.2885).
8. The 1:1 negative-to-positive ratio is inadequate for realistic verification scenarios.
9. Missing critical experimental components: dataset statistics, proper baselines, ablation studies, and per-digit analysis.

**Questions:**

1. Can the authors clearly define whether the task is identity verification (confirming who is speaking) or content verification (confirming what was said), and reframe the paper accordingly?
2. How would results change under a speaker-disjoint evaluation protocol where test speakers are unseen during training?
3. What is the justification for using audio ASR boundaries in a system motivated for visual-only verification? How sensitive are results to boundary accuracy?
4. How does the proposed approach meaningfully differ from or improve upon Lan et al. [5]'s landmark-based features?
5. Can the authors add a DTW-based baseline to justify the time series framing of the contribution?
6. What happens to performance with more realistic negative ratios (e.g., 1:10 or 1:100)?

---

### Official Review · Reviewer_1Dr2 · 2026-05-18

**Rating:** 5
**Confidence:** 3

**Summary:**

The paper proposes a system for verifying whether a speaker's lip motion matches a claimed sequence of Chinese digits. Five geometric features (vertical aperture, horizontal spread, mouth area, roundness, speed) are extracted from facial landmarks via MediaPipe, then fed into a 1D-CNN + BiGRU encoder paired with a digit embedding tower. A baseline system achieves AUC=0.777, EER=0.289 on a Chinese digit dataset.

**Strengths:**

1. **The inter-ocular distance normalization is a practical strength.** Dividing features by inter-ocular distance provides built-in speaker and distance invariance. This is a simple but effective technique that could enable cross-speaker generalization without explicit normalization training. The proposal should highlight and test this specifically (e.g., speaker-disjoint evaluation where this normalization is the key enabler).

2. **The co-articulation illustration (Figure 3) shows genuine domain understanding.** Visualizing how adjacent digits deform lip trajectories demonstrates awareness of the hardest aspect of continuous speech verification. This is the right challenge to focus on.

3. **Complete end-to-end pipeline.** From raw video through landmark detection, feature extraction, temporal encoding, to verification output -- all components are present and functional.

**Weaknesses:**

1. **Chinese digit-specific challenges are underexplored.** Several Mandarin digits share nearly identical outer lip shapes: "yi" (一), "qi" (七), and "si" (四) all involve similar narrow-mouth articulations. The 5-feature representation, which captures only outer lip contour geometry, likely cannot discriminate these. A per-digit breakdown of verification accuracy (a confusion matrix) is essential to understand whether the system works on easy digits (e.g., "ba" 八 with wide mouth opening) but fails on confusable ones.

2. **MediaPipe landmark reliability under real conditions is not discussed.** MediaPipe face mesh assumes frontal faces with reasonable lighting and minimal occlusion. In practice, head rotation during speech, partial lip occlusion (hand gestures), and variable lighting cause landmark jitter and dropped frames. The proposal should discuss: (a) what happens when landmarks are unreliable for some frames, and (b) how the temporal model handles missing or noisy frames. A simple robustness test (add Gaussian noise to landmarks, measure accuracy degradation) would address this.

3. **Speaking rate variability is unaddressed.** The same digit sequence spoken at different speeds produces lip motion trajectories of different lengths and different dynamics. The BiGRU handles variable-length input, but the 5-feature magnitudes (especially speed f5) are rate-dependent. No temporal normalization or rate-invariant feature design is discussed. Consider normalizing by sequence duration or using rate-independent features (e.g., shape ratios rather than raw speed).

4. **The practical use case for lip-text *content* verification is unclear.** If the system verifies that lip motion matches a claimed text, the deployment scenario must specify who provides the "claimed text" and why. Is this for liveness detection (prove you're speaking live by matching a random prompt)? For audio-visual diarization? For detecting deepfake videos? The proposal should pick one scenario and design the evaluation around it. For liveness detection, the "claimed text" would be a randomly generated challenge -- this connects naturally to the security angle without the identity verification confusion.

**Questions:**

1. Which Mandarin digit pairs are most confusable under your 5-feature representation? Can you share a per-digit or per-pair accuracy breakdown?
2. How robust is MediaPipe landmark detection across head poses and lighting conditions in your dataset? Any frames dropped?
3. Does your system handle variable speaking rates? Have you tested with fast vs. slow speakers?
4. What is the intended deployment scenario -- liveness detection, deepfake detection, or something else? This would clarify the evaluation design.

---

### Official Review · Reviewer_1Dr2 · 2026-05-18

**Rating:** 5
**Confidence:** 3

**Summary:**

The paper proposes a system for verifying whether a speaker's lip motion matches a claimed sequence of Chinese digits. Five geometric features (vertical aperture, horizontal spread, mouth area, roundness, speed) are extracted from facial landmarks via MediaPipe, then fed into a 1D-CNN + BiGRU encoder paired with a digit embedding tower. A baseline system achieves AUC=0.777, EER=0.289 on a Chinese digit dataset.

**Strengths:**

1. **The inter-ocular distance normalization is a practical strength.** Dividing features by inter-ocular distance provides built-in speaker and distance invariance. This is a simple but effective technique that could enable cross-speaker generalization without explicit normalization training. The proposal should highlight and test this specifically (e.g., speaker-disjoint evaluation where this normalization is the key enabler).

2. **The co-articulation illustration (Figure 3) shows genuine domain understanding.** Visualizing how adjacent digits deform lip trajectories demonstrates awareness of the hardest aspect of continuous speech verification. This is the right challenge to focus on.

3. **Complete end-to-end pipeline.** From raw video through landmark detection, feature extraction, temporal encoding, to verification output -- all components are present and functional.

**Weaknesses:**

1. **Chinese digit-specific challenges are underexplored.** Several Mandarin digits share nearly identical outer lip shapes: "yi" (一), "qi" (七), and "si" (四) all involve similar narrow-mouth articulations. The 5-feature representation, which captures only outer lip contour geometry, likely cannot discriminate these. A per-digit breakdown of verification accuracy (a confusion matrix) is essential to understand whether the system works on easy digits (e.g., "ba" 八 with wide mouth opening) but fails on confusable ones.

2. **MediaPipe landmark reliability under real conditions is not discussed.** MediaPipe face mesh assumes frontal faces with reasonable lighting and minimal occlusion. In practice, head rotation during speech, partial lip occlusion (hand gestures), and variable lighting cause landmark jitter and dropped frames. The proposal should discuss: (a) what happens when landmarks are unreliable for some frames, and (b) how the temporal model handles missing or noisy frames. A simple robustness test (add Gaussian noise to landmarks, measure accuracy degradation) would address this.

3. **Speaking rate variability is unaddressed.** The same digit sequence spoken at different speeds produces lip motion trajectories of different lengths and different dynamics. The BiGRU handles variable-length input, but the 5-feature magnitudes (especially speed f5) are rate-dependent. No temporal normalization or rate-invariant feature design is discussed. Consider normalizing by sequence duration or using rate-independent features (e.g., shape ratios rather than raw speed).

4. **The practical use case for lip-text *content* verification is unclear.** If the system verifies that lip motion matches a claimed text, the deployment scenario must specify who provides the "claimed text" and why. Is this for liveness detection (prove you're speaking live by matching a random prompt)? For audio-visual diarization? For detecting deepfake videos? The proposal should pick one scenario and design the evaluation around it. For liveness detection, the "claimed text" would be a randomly generated challenge -- this connects naturally to the security angle without the identity verification confusion.

**Questions:**

1. Which Mandarin digit pairs are most confusable under your 5-feature representation? Can you share a per-digit or per-pair accuracy breakdown?
2. How robust is MediaPipe landmark detection across head poses and lighting conditions in your dataset? Any frames dropped?
3. Does your system handle variable speaking rates? Have you tested with fast vs. slow speakers?
4. What is the intended deployment scenario -- liveness detection, deepfake detection, or something else? This would clarify the evaluation design.

---

### Official Review · Reviewer_oT4R · 2026-05-19

**Rating:** 8
**Confidence:** 3

**Summary:**

This proposal presents a lip-text verification system using compact geometric lip-motion features extracted from facial landmarks. The method represents each video as a multivariate time series of five normalized lip features, then uses a hybrid 1D-CNN + BiGRU model to verify whether the observed lip motion matches a claimed digit sequence.

**Strengths:**

The project is well motivated by practical security use cases such as liveness detection, identity verification, and audio-visual consistency checking. The feature design is clear and interpretable, using mouth aperture, horizontal spread, inner lip area, compactness, and lip speed. The proposal also includes preliminary results, with AUC 0.777, EER 0.2885, and accuracy 0.712, showing that the approach is feasible.

**Weaknesses:**

The dataset and evaluation protocol are not described in enough detail. It is unclear how many speakers, videos, and train/test identities are used, which makes it difficult to judge generalization. The negative-pair construction may also be too artificial, since shuffled or replaced digits may not fully represent realistic spoofing or mismatch attacks. The method also relies heavily on landmark quality, but robustness to tracking errors, pose changes, lighting, and phone-camera variation is not deeply evaluated.

**Questions:**

How many speakers and videos are included in the dataset?
Is the train/test split speaker-independent?
How realistic are the generated negative pairs compared to real spoofing or audio-video mismatch attacks?
How robust is the method to landmark detection errors, head pose variation, or poor lighting?
Would a stronger baseline, such as raw video CNN or LipNet-style model, be included for comparison?